# Single vesicle analysis reveals the release of tetraspanin positive extracellular vesicles into circulation with high intensity intermittent exercise

Luke C. McIlvenna[1,2], Hannah-Jade Parker[1,3], Alex P. Seabright[1], Benedict Sale[1], Genevieve Anghileri[1,4], Samuel R.C. Weaver[1], Samuel J.E. Lucas[1] 🆔 and Martin Whitham[1,3] 🆔

[1]*School of Sport, Exercise and Rehabilitation Sciences, University of Birmingham, Birmingham, UK*
[2]*Epigenetics & Cellular Senescence Group, Blizard Institute, Barts and the London School of Medicine and Dentistry, Queen Mary University of London, London, UK*
[3]*MRC-Versus Arthritis Centre for Musculoskeletal Ageing Research, University of Birmingham, Birmingham, UK*
[4]*School of Sport, Exercise and Health Sciences, Loughborough University, Loughborough, UK*

Handling Editors: Harold Schultz & Susan Currie

The peer review history is available in the Supporting Information section of this article (https://doi.org/10.1113/JP284047#support-information-section).

**Abstract**  Small extracellular vesicles (sEVs) are released from all cell types and participate in the intercellular exchange of proteins, lipids, metabolites and nucleic acids. Proteomic, flow cytometry and nanoparticle tracking analyses suggest sEVs are released into circulation with exercise. However, interpretation of these data may be influenced by sources of bias introduced by different analytical

L. C. McIlvenna and H.-J. Parker contributed equally to this work.

The Journal of Physiology

approaches. Seven healthy participants carried out a high intensity intermittent training (HIIT) cycle protocol consisting of $4 \times 30$ s at a work-rate corresponding to 200% of individual max power (watts) interspersed by 4.5 min of active recovery. EDTA-treated blood was collected before and immediately after the final effort. Platelet-poor (PPP) and platelet-free (PFP) plasma was derived by one or two centrifugal spins at 2500 $g$, respectively (15 min, room temperature). Platelets were counted on an automated haemocytometer. Plasma samples were assessed with the Exoview R100 platform, which immobilises sEVs expressing common tetraspanin markers CD9, CD63, CD81 and CD41a on microfluidic chips and with the aid of fluorescence imaging, counts their abundance at a single sEV resolution, importantly, without a pre-isolation step. There was a lower number of platelets in the PFP than PPP, which was associated with a lower number of CD9, CD63 and CD41a positive sEVs. HIIT induced an increase in fluorescence counts in CD9, CD63 and CD81 positive sEVs in both PPP and PFP. These data support the concept that sEVs are released into circulation with exercise. Furthermore, platelet-free plasma is the preferred, representative analyte to study sEV dynamics and phenotype during exercise.

(Received 1 November 2022; accepted after revision 22 February 2023; first published online 28 February 2023)

**Corresponding author** M. Whitham: School of Sport, Exercise and Rehabilitation Sciences, University of Birmingham, Edgbaston, Birmingham, B15 2TT, UK. Email: m.whitham@bham.ac.uk

**Abstract figure legend** Platelet-free plasma was derived from seven healthy participants before and after a high intensity intermittent training (HIIT) exercise protocol. Samples were directly analysed via a microfluidic chip array, which immobilises small extracellular vesicles (sEV) expressing the tetraspanin protein markers CD9, CD63, CD81 and CD41a and determines sEV count and protein expression via fluorescence intensity on a single-sEV basis. HIIT resulted in an increase in the number of CD9, CD63, CD81 and CD41a positive sEVs in circulation, with an associated increase in CD9, CD63 and CD81 protein expression. Since platelets are known to release sEVs, also analysed were sEV counts in platelet-free *versus* platelet-poor plasma. Since there was a significant reduction in CD9, CD63 and CD41a positive sEVs associated with a decrease in platelets, platelet-free plasma is likely the most representative analyte when examining sEV dynamics during exercise.

## Key points

- Small extracellular vesicles (sEV) are nano-sized particles containing protein, metabolites, lipid and RNA that can be transferred from cell to cell.
- Previous findings implicate that sEVs are released into circulation with exhaustive, aerobic exercise, but since there is no gold standard method to isolate sEVs, these findings may be subject to bias introduced by different approaches.
- Here, we use a novel method to immobilise and image sEVs, at single-vesicle resolution, to show sEVs are released into circulation with high intensity intermittent exercise.
- Since platelet depletion of plasma results in a reduction in sEVs, platelet-free plasma is the preferred analyte to examine sEV dynamics and phenotype in the context of exercise.

## Introduction

Extracellular vesicles (EV) are lipid bilayer-encased particles known to participate in the exchange of molecules from cell to cell (van Niel et al., 2018). In focus here are EVs of endosomal origin, often referred to as exosomes, and EVs shed from the plasma membrane, or 'microparticles'. In the absence of an established means to clearly distinguish each type, we use here the umbrella term, 'small extracellular vesicles' (sEV) constituting extracellular vesicles of <200 nm in size

(Thery et al., 2018). Importantly, analyses of the molecular contents of sEVs in various biological scenarios reveal vast complexity, creating intrigue as to the fundamental biological role of sEVs in these contexts.

Of late, studies have implied, with some exceptions (Rigamonti et al., 2019; Watanabe et al., 2022), that there is a transient release of sEVs into circulation in response to the whole-body stress elicited by an acute bout of exercise in healthy human participants (reviewed in Denham & Spencer, 2020). This hypothesis appears to be supported by experimental approaches that aimed

to quantify changes, during exercise, in protein markers (Brahmer et al., 2019; Fruhbeis et al., 2015; Guescini et al., 2015; Helmig et al., 2015; Just et al., 2020) or microRNAs (D'souza et al., 2018; Guescini et al., 2015; Just et al., 2020; Lovett et al., 2018) assumed to be reflective of sEVs. Furthermore, we and others (Vanderboom et al., 2021; Whitham et al., 2018) have carried out unbiased, high coverage, shotgun proteomic analyses of crude EV isolates from platelet-free plasma taken from exercising human participants. In these analyses, despite injection of the same amount of peptide from resting and post-exercise states, there emerges a robust increase of several classes of protein known to constitute sEVs, implying a considerable release of these vesicles into circulation. In further support, experiments carrying out nano-particle tracking analyses have shown quantitative increases in nano-sized particles in response to exercise (Fruhbeis et al., 2015; Oliveira et al., 2018; Whitham et al., 2018), although interpretation of these data may be limited by the known contamination of lipoprotein particles in these analyses (Brahmer et al., 2020; Gardiner et al., 2016). Collectively, these investigations stimulate speculation that sEVs may facilitate the exchange of secreted factors between cells and tissues, perhaps to orchestrate a complex molecular response to the metabolic and cardiorespiratory demands of exercise.

However, much of the work to date may have been subject to bias by the selected method of sEV isolation and characterization and the lack of a universal, 'gold standard' approach. It is well known that analysis of sEVs can be heavily impacted by how they were separated from other constituents of complex analytes like plasma. When choosing which isolation method to adopt, researchers are generally challenged with a trade-off between sEV recovery and specificity, with higher yield isolation methods tending to exhibit contamination with non-EV co-isolates and higher specificity isolation methods yielding little material to work with (Coccoza et al., 2020). Research to date has adopted a range of isolation methods, such as polymer-based precipitation, ultrafiltration, differential centrifugation and size exclusion chromatography, emphasising the lack of a gold standard method to purify sEVs of interest.

Additionally, many of these methods involve a pooled analysis of EVs and do not allow for an understanding of whether specific sEV proteins are expressed universally or to distinguish if they are located on the membrane or in the lumen of the vesicle.

By way of an attempt to overcome any potential isolation method bias that might contribute to interpretation of sEV responses to exercise, we have directly analysed plasma samples from exercising human participants using a single particle, immunocapture fluorescence-based analysis (Deng et al., 2022). Here, plasma samples, in very small volumes, are loaded onto microfluidic chips, importantly without a pre-isolation step, whereby sEVs positive for tetraspanin proteins CD9, CD63, CD81 and CD41a are immobilised. This facilitates a quantitative appreciation of sEVs via amplification of the signal by additional probing with different coloured fluorescence antibodies to CD9, CD63 and CD81. It is hypothesised that a high intensity intermittent training (HIIT) exercise protocol will induce an increase in the counts of CD9, CD63 and CD81 positive sEVs.

An additional source of bias in studies examining sEV responses to exercise is the chosen blood handling procedure. In particular, studies imply remnant platelets, unsuccessfully removed from plasma during sample processing, can release sEVs *ex vivo* (Karimi et al., 2022). Therefore, as an added aim, we carried out a direct assessment of sEV counts in platelet-poor and platelet-free plasma to investigate the most representative analyte to use in studies examining sEVs in the context of exercise.

## Methods

### Ethical approval

Ethical approval was obtained from the University of Birmingham Human Research Ethics Committee (Reference ERN_17-1570), and all participants provided written informed consent prior to participation in the study. The study conformed to the standards set by the *Declaration of Helsinki*, except for registration in a database.

**Luke C. McIlvenna** is a postdoctoral researcher at Queen Mary University of London at the Blizzard institute in the Epigenetics & Cellular Senescence lab. He is investigating the role of extracellular vesicles in cellular senescence and rejuvenation. He obtained his PhD from Victoria University, Melbourne in 2021 and carried out this work in his first postdoctoral position in the Tissue Cross-Talk laboratory (2021–2022). **Hannah-Jade Parker** is a PhD student in the School of Sport, Exercise and Rehabilitation Sciences at the University of Birmingham. After completing an undergraduate degree in Biology, she completed an MSc in Medical Biosciences at the University of Bath followed by a stint in industry in the immuno-oncology field. In 2019, she began her doctoral studies with a studentship with the Medical Research Council Versus Arthritis Centre for Musculoskeletal Ageing Research post-graduate training programme.
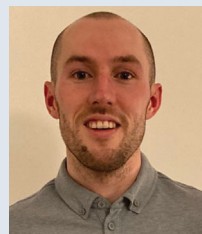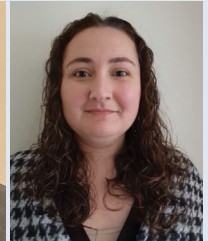

## Participants and experimental procedures

Seven healthy, recreationally active participants (2 females, 5 males, age 21 ± 4 years, body mass index 23 ± 2 kg/m$^2$, $\dot{V}_{O_2peak}$ 44.4 ± 5.2 ml/kg/min) were recruited. Each participant was required to refrain from consuming a large meal <4 h, and from consuming any caffeine <6 h prior to each laboratory session. Participants were encouraged to drink at least 0.5 litres of water within 4 h of the study, and to consume 0.25 litres within 15 min of each activity to prevent dehydration. To determine maximal exercise capacity, a standardised, incremental aerobic cycling test (Lode ergometer, Groningen, Netherlands) was carried out, consisting of a 5 min warm-up at 50 W, followed by a ramped protocol by which loaded resistance increased every 3 min until volitional exhaustion. Inspired and expired gases were assessed throughout, using a Vintus CPX breath-by-breath metabolic cart (Vyaire Medical, Chineham, UK). From this initial test, $\dot{V}_{O_2peak}$ and maximum power ($P_{Max}$; watts) were determined and a subsequent work rate relative to $P_{Max}$ was calculated for the subsequent high intensity exercise session (HIIT). HIIT sessions for each participant took place between 2 and 14 days after the initial testing, consisting of a 5 min warm-up period, followed by 4 × 30 s of cycling at 200% $P_{Max}$, with 4 min and 30 s of active recovery at 50 W between each effort (Fig. 2*A*). This was followed by a 5 min cool-down. Heart rate was measured continuously using a chest transmitter and receiver (Polar Electro Oy, Kempele, Finland). All exercise tests were carried out at the same time of day.

## Blood collection and handling

**Blood collection.** Whole blood samples were collected from an in-dwelling catheter inserted into an antecubital vein and kept patent with frequent infusion of saline. For each let, 2 ml of venous blood was drawn and disposed of before 20 ml of blood was drawn into an EDTA coated tube (BD, Eysins, Switzerland). Experimental samples were collected from each participant after 10 min of supine resting (REST) and within 30 s of cessation of the HIIT protocol (see Fig. 2*A*).

**Platelet-poor plasma and platelet-free plasma.** For the assessment of the effect of remnant platelets on sEV counts, blood samples ($n = 9$) were spun (Sorval Legend X1, Thermo Fisher Scientific, Waltham, MA, USA) within 2 min of sampling. For platelet-poor plasma (PPP), blood samples were subjected to a single centrifugation at 2500 *g* for 15 min at room temperature (see Fig. 1*A*). Plasma was removed, leaving ∼50 mm space to avoid the buffy coat and placed into a separate tube; 2 ml of this plasma was collected and frozen at −80°C. The remaining PP plasma

was spun again at 2500 *g* for 15 min at room temperature. Avoiding the bottom 50 mm, supernatant was placed into a separate tube for platelet-free plasma (PFP) and stored at −80°C.

## Platelet counts

Platelet counts were obtained from platelet-poor and platelet-free plasma using an automated haematology analyser (Yumizen H500, Horiba Medical). A total of 20 $\mu$l was used for the analysis.

## Single EV particle analysis

ExoView human tetraspanin plasma Kits (EV-TETRA-P) were used according to the manufacturer's instructions (Unchained Laboratories, Pleasanton, CA, USA). Briefly, for the incubation step, microfluidic chips were placed in the centre of a 24-well plate. The microfluidic chips are coated with individual antibody capture spots for CD9, CD63, CD81, CD41a and mouse IgG as a negative control. Then, 10 $\mu$l of plasma was diluted in 990 $\mu$l of the incubation buffer. Subsequently, 35 $\mu$l of the diluted sample was pipetted on the centre of each chip and the plate was sealed with an adhesive cover and incubated overnight at room temperature. The next day, the chips were washed, and staining for immunofluorescence was carried out at antibody dilutions of 1:500 for CD9 (CF488), CD63 (CF647) and CD81 (CF555). Data were analysed using ExoView Analyzer 3.0 software. By and large, we opted to use fluorescence to determine sEV particle counts rather than single particle interferometric reflectance imaging sensing (SP-IRIS), based upon previous work showing the fluorescence-mode yields a greater sensitivity (Khan et al., 2021; Mizenko et al., 2021). Following correction to negative IgG controls, we obtained the following outcome measures: fluorescence counts, which represents the number of tetraspanin protein positive sEVs in each plasma sample; mean fluorescence intensity (MFI), which represents the protein expression of tetraspanins on sEVs; and size of the captured sEVs on each tetraspanin-specific spot.

## Follow-up assessment

In an additional follow-up to the main study, five participants (3 females, 2 males, age 21 ± 3 years, $\dot{V}_{O_2peak}$ 41.0 ± 4.4 ml/kg/min) carried out the same HIIT protocol (see Fig. 2*A*). Single spun, pre and post plasma samples were analysed for tetraspanin positive EV counts and protein expression as described above.

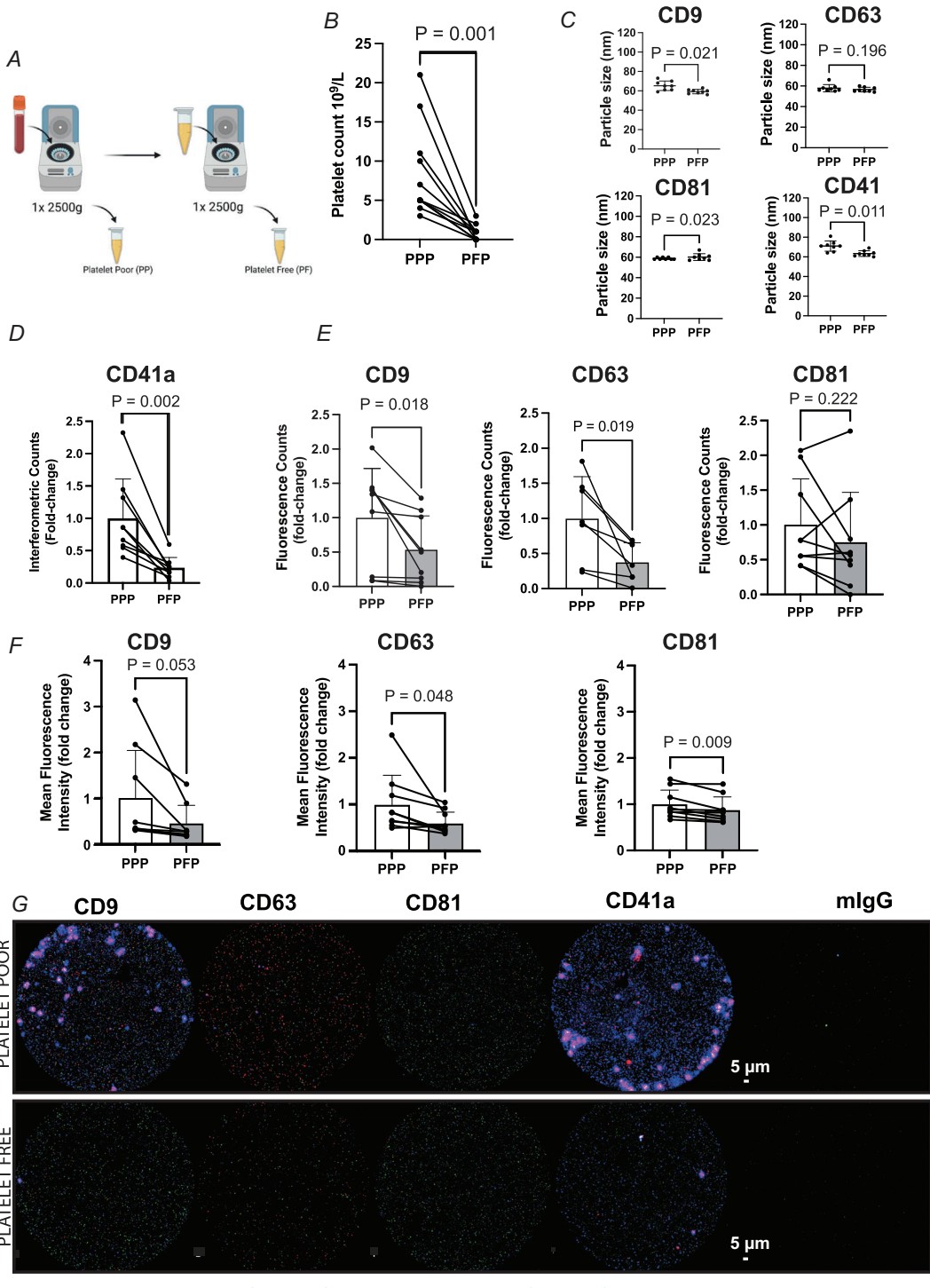

**Figure 1. Platelet depletion influences basal tetraspanin positive small extracellular vesicles profile**

*A* and *B*, illustration of platelet removal protocol (*A*) and subsequent platelet counts in platelet-poor (PPP) and platelet-free (PFP) plasma (*B*). *C*, sEV particle size in PPP *vs*. PFP. *D*, SP-IRIS assessment of CD41a$^+$ sEV number in PPP *vs*. PFP. *E*, the number of plasma tetraspanin positive sEVs in PPP *vs*. PFP represented by the summed fluorescence counts of CD9, CD63 or CD81 fluorescent antibodies across all capture spots. *F*, tetraspanin protein expression in PPP *vs*. PFP represented by summed mean fluorescence intensity of CD9, CD63 and CD81 antibodies across all capture spots. *G*, representative images of CD9 (CF488), CD63 (CF647), and CD81 (CF555) fluorescence signalling for each capture spot. *n* = 9, except for CD63 counts (*n* = 7), since two participants' data were outside the range of detection. *P*-values for fold change were determined with Student's paired *t* test. [Colour figure can be viewed at wileyonlinelibrary.com]

## Statistical analyses

All analysis and data visualisation was carried out using GraphPad Prism version 9 (GraphPad Software Inc., San Diego, CA, USA). Comparisons for fluorescence counts, mean fluorescence intensity, platelet counts and sEV size were determined with a two-tailed paired samples Student's *t* test. Given the modest *n*, Cohen's *d* was calculated to estimate the size of the effect where appropriate. Heart rate data were analysed using a one-way ANOVA with post-hoc analyses calculated by Dunnett's multiple comparison test. Data are reported as means ± standard deviation (SD) unless stated otherwise, and statistical significance was declared when $P < 0.05$.

## Results

### Influence of platelet depletion on the basal tetraspanin profile of plasma extracellular vesicles

Platelets are known to be a source of sEVs (Nolan & Jones, 2017). Therefore, sample handling may have implications for the number of platelets in biological samples and confound observed changes in sEV profile in response to given stimuli. As expected, we observed a reduction in the number of platelets when comparing plasma from a single centrifugation (i.e. PPP) *versus* double centrifugation (i.e. PFP) step (Fig. 1*B*) ($P = 0.001$, platelet poor: $(8.45 \pm 5.82) \times 10^9$/l and platelet free: $(0.82 \pm 0.98) \times 10^9$/l). Platelet removal resulted in

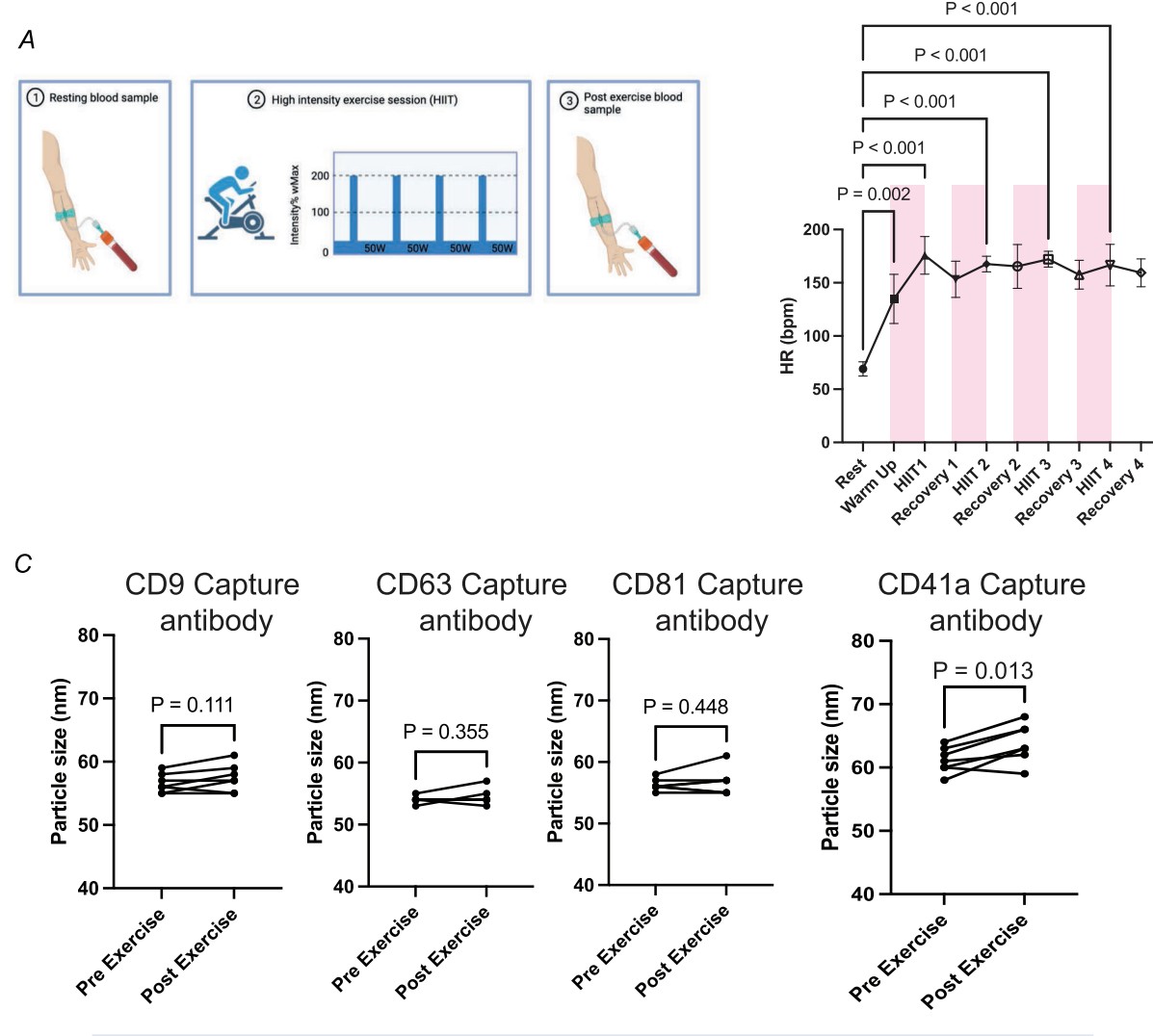

**Figure 2. High intensity intermittent training (HIIT) exercise protocol**
*A*, schematic representation of performed exercise protocol and blood sampling. *B*, exercise induced a significant elevation in heart rate from rest following each effort (one-way ANOVA and Dunnett's multiple comparison test). *C*, sEV particle size pre- and post-exercise in CD9, CD63, CD81 and CD41a positive particles (*n* = 7). *P*-values for fold change were determined with Student's paired *t* test. [Colour figure can be viewed at wileyonlinelibrary.com]

a small reduction in CD9$^+$ and CD41a$^+$ particle size (Fig. 1*C*). Importantly, the drop in platelet count was associated with an alteration in sEV count and tetraspanin protein expression in the analysed plasma samples. In the absence of a CD41a fluorescent antibody, SP-IRIS measurements of sEVs captured on the CD41a spot revealed a reduction in the number of sEVs positive for this commonly used platelet marker ($P = 0.002$; Fig. 1*D*). Furthermore, we observed a reduction in the number of CD9 ($P = 0.018$) and CD63 ($P = 0.019$) positive sEVs, while CD81$^+$ sEVs remained unchanged (Fig. 1*E*). Total protein expression was also reduced for CD63 ($P = 0.048$) and CD81 ($P = 0.009$, Fig. 1*F*). Representative fluorescence images for these analyses are depicted in Fig. 1*G*. Assuming tetraspanins CD9, CD63, CD81 and CD41a are appropriate capture targets for sEVs, a drop in sEV count with a reduction in platelets in plasma supports the growing consensus that double spun, 'platelet-free' plasma is the most representative analyte for circulating sEVs.

### Effects of high intensity intermittent exercise on the number of circulating tetraspanin positive small extracellular vesicles

Platelet-free plasma samples from exercising volunteers were subsequently analysed using the same immuno-capture/imaging system (Fig. 2*A*). The HIIT protocol induced a significant degree of exertion in all participants, as indicated by a significant increase in heart rate after each bout to near maximal levels (Fig. 2*B*). The exercise protocol did not change the vesicle size appearing in circulation in CD9, CD63 and CD81 captured sEVs. There was a small increase in particle size in CD41a captured sEVs with exercise (Fig. 2*C*).

With respect to sEV number, the fluorescence counts increased following HIIT for CD9 ($P = 0.018$, $d = 0.86$), CD63 ($P = 0.032$, $d = 1.12$) and CD81 ($P = 0.041$, $d = 0.29$; Fig. 3*A*), with SP-IRS analysis also showing an increase in CD41a positive sEVs ($P = 0.013$, Fig. 3*B*). The increase in sEV number was accompanied by an increase in CD9 ($P = 0.008$, $d = 2.1$) and CD63 ($P = 0.003$, $d = 1.2$), but not CD81 protein expression following HIIT (Fig. 3*C*).

While these data represent the sum of each fluorescent probe across all capture spots (Fig. 3*C*), colocalization analysis provides a deeper appreciation of the number of sEVs containing one, two or combinations of tetraspanin markers (Fig. 4). Tetraspanin profiles of sEV remained heterogeneous with HIIT (Fig. 4*A*) and the post-exercise increase in counts appeared to be largely driven by sEVs captured by or expressing CD9 uniquely or in combination with the other markers.

Collectively, these data reveal medium to large effect sizes and support previous findings in pre-isolated plasma

samples, that sEVs are released into circulation with exertive exercise.

### Effect of exercise on change in circulating sEV number prevails in single spun plasma samples

Since we observed a robust response to exercise in CD9, CD63 and CD81 positive sEVs, we hypothesised that this effect may prevail in single spun plasma samples, reasoning that the effect of removal of platelets from plasma has a comparable effect on sEV count in both pre- and post-exercise plasma. In an additional cohort of participants carrying out the same HIIT protocol, we observed an increase in CD9 ($P = 0.02$), CD63 ($P = 0.03$) and CD81 ($P = 0.04$) fluorescence counts (Fig. 5*A*) following exercise in single spun (PPP) plasma, accompanied by an increase in interferometric counts of CD41a positive sEVs ($P = 0.031$ (Fig. 5*B*). Mean fluorescence intensity increased in CD9$^+$ sEVs but not CD63$^+$ or CD81$^+$ sEVs (Fig. 5*C*).

### Discussion

It has been proposed that sEVs could play a role in the adaptation to exercise via intracellular and inter-tissue communication (Denham & Spencer, 2020; Safdar & Tarnopolsky, 2018; Whitham et al., 2018). Indeed, sEVs have been demonstrated, albeit predominantly in rodent studies, to exert functional effects influencing recovery from myocardial injury (Bei et al., 2017), immune function (Robins & Morelli, 2014), muscle mass (Wu et al., 2022) and metabolism (Castaño et al., 2018; Castaño et al., 2020). Furthermore, sEVs have been attributed to improvements in insulin sensitivity following 12 weeks of HIIT in human patients (Apostolopoulou et al., 2021). Although more work is required to unpack the functional effects of sEVs *in vivo*, there appears to be sufficient evidence to indicate that they may play a role in the adaptive responses to exercise.

Several groups have demonstrated that a single bout of aerobic exercise is sufficient to induce a transient increase in the number of sEVs in circulation (e.g. Fruhbeis et al., 2015; Vanderboom et al., 2021; Whitham et al., 2018). This increase has been observed when sEVs are isolated using common isolation methods such as ultracentrifugation and size exclusion chromatography which may add elements of bias to the downstream analytical approach. Here, we utilised single EV particle analysis to demonstrate a substantial increase in circulating sEVs following HIIT. A key advantage to this approach is that it requires no pre-isolation step, conceivably reducing the bias and confounders that may be introduced by EV isolation methods. We observed an increase in the number of CD9, CD63 and CD81 sEVs

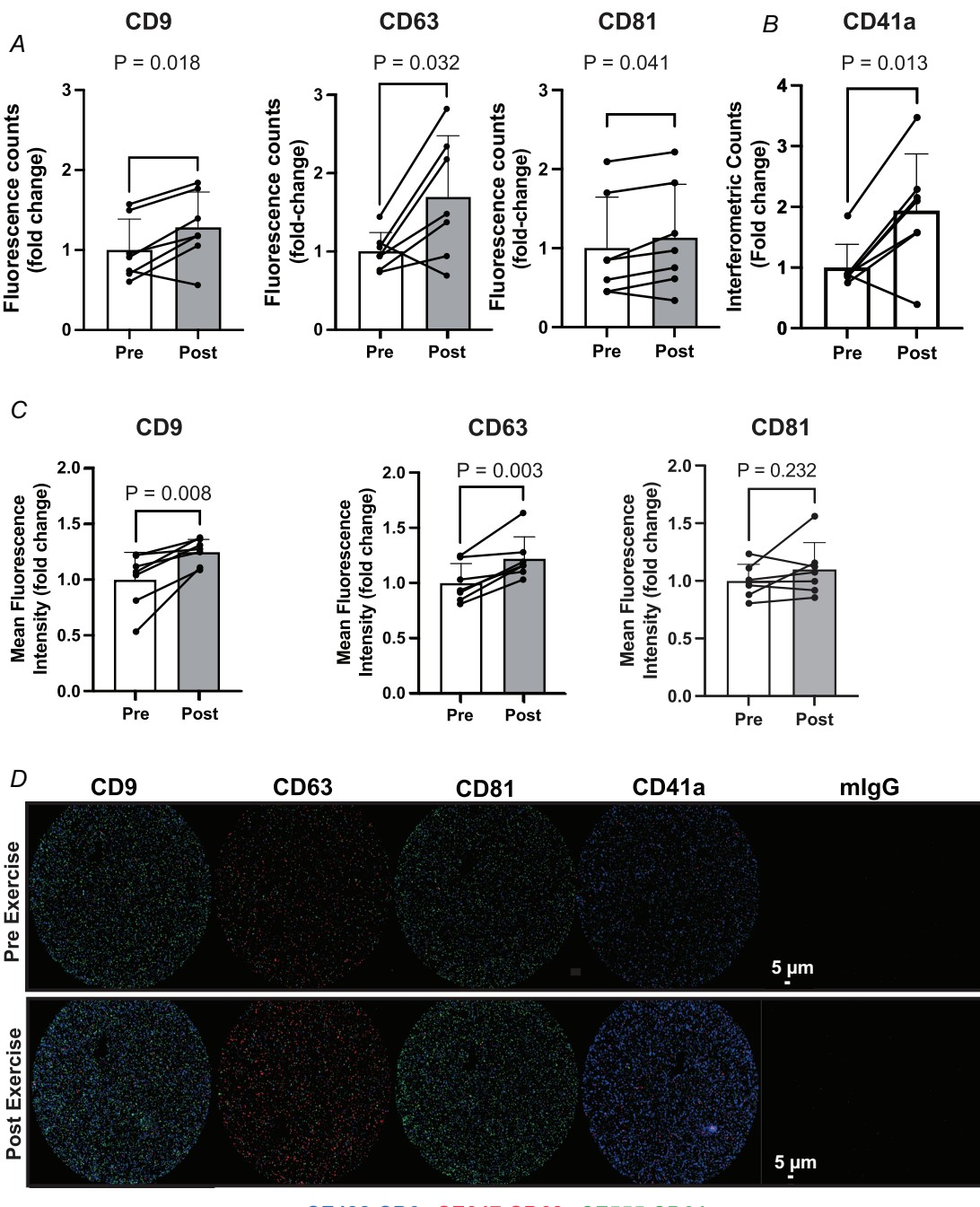

**Figure 3. HIIT exercise induces an increase in the number of circulating tetraspanin positive small extracellular vesicles and alters tetraspanin expression**

*A*, the number of plasma tetraspanin positive sEVs in pre- and post-exercise, platelet-free plasma represented by the summed fluorescence counts of CD9, CD63 or CD81 fluorescent antibodies across all capture spots. *B*, SP-IRIS assessment of CD41a$^+$ sEV number in pre- and post-exercise. *C*, tetraspanin protein expression in pre- *vs.* post-exercise plasma represented by summed mean fluorescence intensity of CD9, CD63 and CD81 antibodies across all capture spots. *D*, representative images of CD9 (CF488), CD63 (CF647) and CD81 (CF555) fluorescence signalling for each capture spot. *n* = 7. *P*-values for fold change were determined with Student's paired *t* test. [Colour figure can be viewed at wileyonlinelibrary.com]

following exercise, which was accompanied by an increase in the expression of CD9 and CD63. Of note, tetraspanins play a role in sEV uptake and paracrine signalling. For example, silencing or knockdown of CD9 in sEVs and recipient cells reduces sEV uptake (Morelli et al., 2004; Soekmadji et al., 2017; Santos et al., 2019; Nigri et al., 2022). The increase in the number of $CD9^+$ particles and surface expression observed in our study could represent a mechanism by which exercise promotes not only sEV release but also their uptake.

A challenge for researchers working in the field of EVs is the sample volume required for informative analysis and characterization. This is particularly pertinent when dealing with human samples, especially in clinical settings, where limited material is often available. Similar issues can also be faced when working with animal models. Single EV analysis technologies provide a novel approach to overcome this limitation. In the case of our study only 10 $\mu$l of plasma was required for the analysis, ultimately

making the study of EV dynamics in different physiological contexts highly feasible.

The core tetraspanins (CD9, CD63 and CD81) are commonly employed as global EV markers, although it is now becoming more apparent that their expression is highly variable in sEVs derived from different cell types (Garcia-Martin et al., 2022; Kugeratski et al., 2021). This should be taken into consideration when interpreting the results of our study and that the changes observed are possibly representative of a sub-population of sEVs. This reinforces the need for the identification and application of new putative surface sEV markers; for instance ATP1A1, BSG, SLC1A5, SLC3A2, ITGB1 and LGALS3BP have been identified as proteins consistently expressed on the surface of EVs from multiple cell types (Kugeratski et al., 2021). Crucially, immunocapture imaging technologies such as those used here allow for customization with different capture and quantification antibodies. In addition, an underlying assumption in the

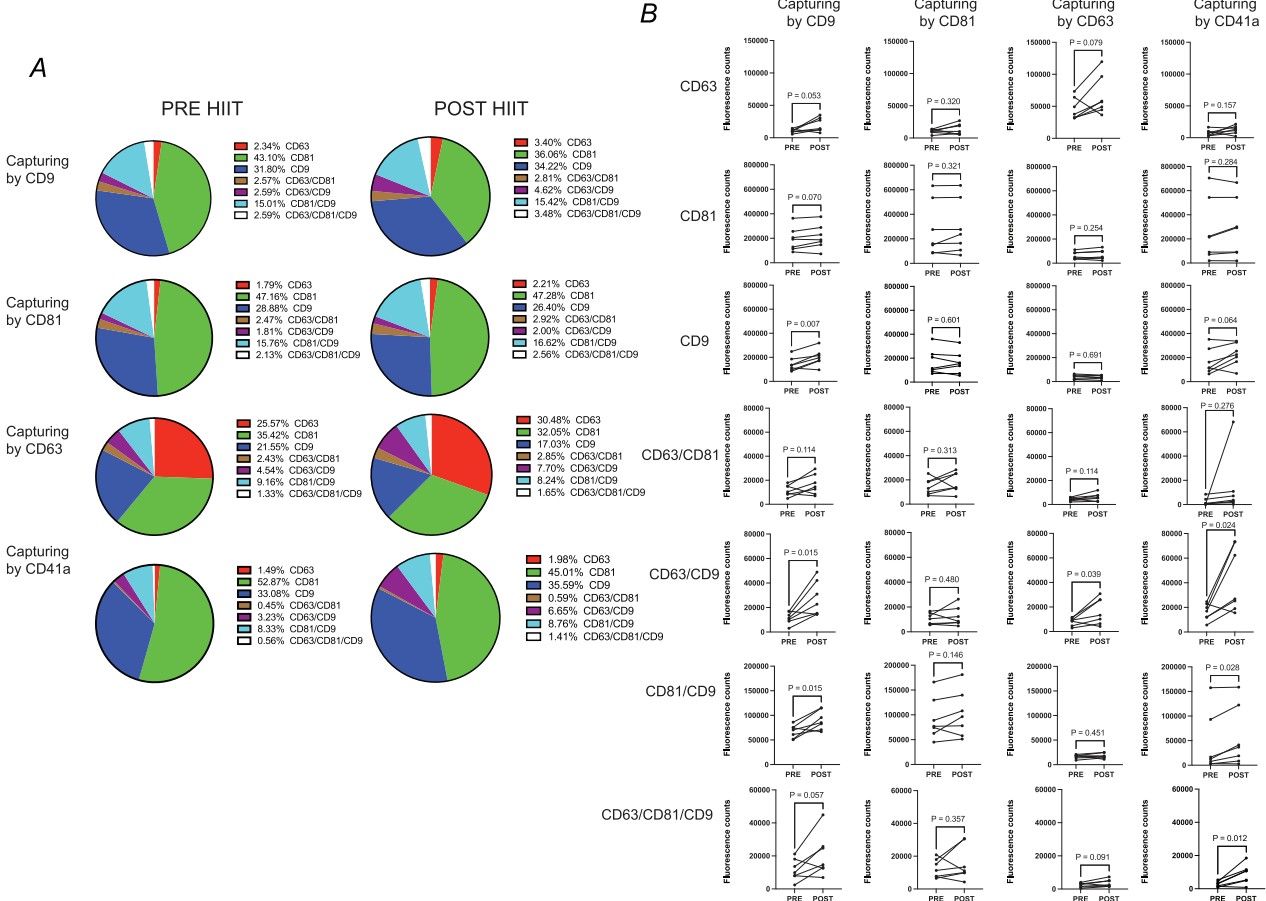

**Figure 4. Colocalisation of tetraspanin protein markers in platelet-free, pre- and post-HIIT plasma**
*A*, the percentage of sEVs expressing each tetraspanin marker, uniquely and in combination, contributing to the total sEV pool captured on each spot. *B*, fluorescence counts for CD9, CD63, CD81 and all possible combinations of each on specific CD9, CD63, CD81 and CD41a capture spots. *n* = 7. *P*-values for pre- *vs*. post-HIIT exercise were determined with Student's paired *t* test. [Colour figure can be viewed at wileyonlinelibrary.com]

presented finding is that the core tetraspanins we have analysed are on sEVs and not released into circulation independently. Since none of these proteins possess a signal peptide, this is unlikely, and a multitude of previous works note a lack of presence of these proteins in the non-EV fraction (Jeppesen et al., 2019; Yoshida et al., 2019; Zhang et al., 2020).

We report here, albeit with a modest sample size, medium to large effect sizes in the sEV response to HIIT exercise. Importantly, only a small number of female participants were included in this analysis, which is an acknowledged limitation, particularly in light of some data suggesting that some vesicle types may respond to exercise differently in male and females (Conkright et al., 2022; Rigamonti et al., 2019; Shill et al., 2018).

An unresolved issue in the field of EVs is the lack of cell source specific sEV markers, meaning that the source of sEVs in circulation and those released during exercise remains speculative. Several attempts have been made to identify cell and tissue specific markers (Garcia-Martin et al., 2022; Vanderboom et al., 2021), but very few proteins or genes are solely expressed in a single cell type or tissue (Reickmann et al., 2017). The lack of specific markers combined with the heterogeneity of sEVs continues to pose a significant challenge for the field. Furthermore, since this approach focuses largely on CD9, CD63 and CD81, widely enriched tetraspanin proteins, it does not offer a potential resolution to this issue. That said, SP-IRS analysis of the platelet-free plasma showed an increase in CD41a$^+$ vesicles in response to exercise (Fig. 3*B*). This may support previous work suggesting a contribution of platelets to the circulating pool *in vivo* (Brahmer et al., 2020), although some data suggest CD41a (ITGA2B) protein expression might not be exclusive to thrombocytes (Reickmann et al., 2017). While it is intuitive that skeletal muscle might release sEVs into circulation with aerobic exercise, this is unlikely to be the sole source (Brahmer et al., 2020; Estrada et al., 2022; Watanabe et al., 2022; Whitham et al., 2018).

The presence of platelets in plasma samples is often cited as an experimental confounder in the EV field, due to the ability of platelets to release sEVs upon activation (Karimi et al., 2022). Our findings confirm the effectiveness of an extra centrifuge step to

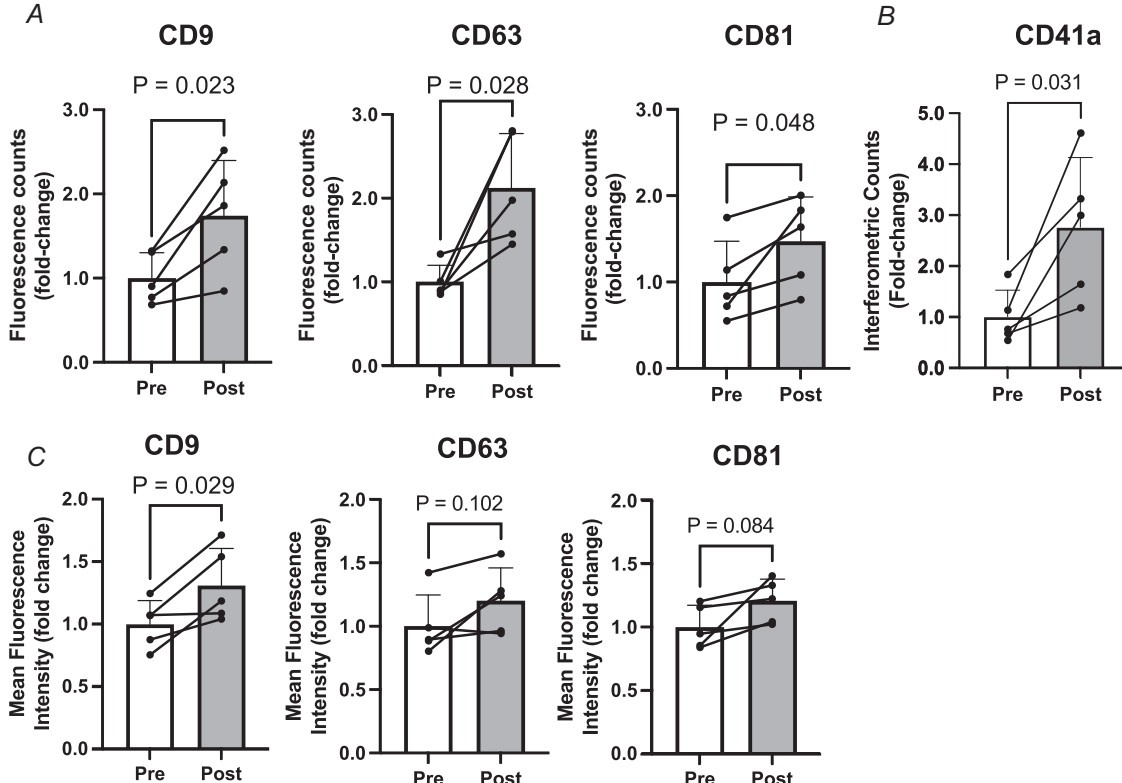

**Figure 5. The fold change effect of HIIT exercise on sEV count prevails in single spun plasma**
*A*, the number of plasma tetraspanin positive sEVs in pre- and post-exercise platelet-poor plasma represented by the summed fluorescence counts of CD9, CD63 or CD81 fluorescent antibodies across all capture spots. *B*, SP-IRIS assessment of CD41a$^+$ sEV number pre- and post-exercise. *C*, tetraspanin protein expression in pre- *vs.* post-exercise plasma represented by summed mean fluorescence intensity of CD9, CD63 and CD81 antibodies across all capture spots. *n* = 5. *P*-values for fold change were determined with Student's paired *t* test.

dramatically reduce the number of platelets present in human plasma samples during initial sample processing. Subsequently we were able to demonstrate that the increased number of plasma sEVs following an acute bout of HIIT can be observed in both platelet-poor (Fig. 3) and platelet-free plasma (Fig. 5). However, we did find that the platelet depletion step influenced the expression of tetraspanins pre- and post-HIIT. When determining the sEV phenotype, the importance of platelet depletion becomes more apparent. This has previously been highlighted in a comparison of the sEV proteome between platelet-poor and platelet-free plasma, whereby significantly more proteins were detected when the depletion step was not employed (Vanderboom et al., 2021). Furthermore, the majority of the additional proteins detected in platelet-poor plasma were related to platelets and leukocytes. This finding is particularly relevant in the context of our data in platelet-poor plasma (Fig. 5), which would seem to indicate an increase in CD41a positive sEV following HIIT exercise. Interpretation of these data is challenging since it is unclear what contribution remnant platelets may have made to the CD41a pool of sEVs *ex vivo*. Likewise, others have shown that a freeze–thaw cycle of platelet-poor plasma results in a substantial increase in EV counts (Artoni et al., 2012). Collectively these data show that the *ex vivo* contribution of platelets can alter both counts and protein composition of EVs.

In addition, it has been noted that residual platelets still remain following double centrifugation (as observed in our study); however platelets can almost be completely removed without affecting the EV yield, by an additional step using a 0.8 $\mu$m polycarbonate filter (Bettin et al., 2022). Future studies should consider employing this rigorous approach to remove the potential *ex vivo* contribution of platelets upon freeze–thaw. Despite the importance of platelet depletion, it has been noted that <5% of EV studies use this approach (Karim et al., 2022), and where possible we would recommend this approach for those who intend to study EV dynamics in the context of exercise.

Overall, we have highlighted the utility of single particle analysis technologies for assessing dynamic changes in sEVs. Using this approach, we were able to confirm previous findings that used alternative methods, with medium to large effects, that sEVs are released into circulation following an acute bout of aerobic exercise. Crucially, this approach did not require sample purification and only required a small volume of plasma (10 $\mu$l). Single EV immunocapture methods therefore provide a sensitive assessment of sEV dynamics in the context of exercise that can complement existing approaches to delineate the biological relevance of sEV release into circulation with exercise. Our results also highlight how platelet depletion, or lack of, can influence the sEV phenotype, reinforcing the importance of consistent sample processing.

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

# Additional information

## Data availability statement

All raw data are available on request to the corresponding author.

## Competing interests

The authors declare no conflict of interest.

## Author contributions

Conceptualisation of the study was provided by M.W. L.C.M. and M.W. carried out the formal analysis. M.W. contributed funding acquisition. Investigation was carried out by all authors, with methodology provided by S.R.C.W., S.J.E.L. and M.W. Project administration was carried out by L.C.M., H.P., S.R.C.W., S.J.E.L. and M.W. Resources and supervision were provided by S.J.E.L. and M.W. Visualisation was carried out by L.C.M. and M.W. Writing of the original document was carried out by L.C.M. and M.W., with review and editing by all authors. All authors have read and approved the final version of this manuscript and agree to be accountable for all aspects of the work in ensuring that questions related to the accuracy or integrity of any part of the work are appropriately investigated and resolved. All persons designated as authors qualify for authorship, and all those who qualify for authorship are listed.

## Funding

This work was funded in part by an award to M.W. from the Wellcome Trust (217 341/Z/19/Z) and a DTP studentship to H.P. from the Medical Research Council Versus Arthritis Centre for Musculoskeletal Ageing Research, University of Birmingham.

## Acknowledgements

The authors would like to thank all participants for taking part in the study, Alex Shephard of Unchained laboratories (Nanoview

Biosciences) and Josh Price of University of Birmingham for assistance with the Exoview analysis.

## Keywords

EV isolation, exercise, secreted factors, small extracellular vesicles, tetraspanins

## Supporting information

Additional supporting information can be found online in the Supporting Information section at the end of the HTML view of the article. Supporting information files available:

**Statistical Summary Document**
**Peer Review History**

