## [Peer Review History · The Journal of Physiology]

Single vesicle analysis reveals the release of tetraspanin positive extracellular vesicles into circulation with high intensity intermittent exercise

Luke C McIlvenna, Hannah-Jade Parker, Alex P Seabright, Benedict Sale, Genevieve Anghileri, Samuel R.C Weaver, Samuel J.E. Lucas, and Martin Whitham

DOI: 10.1113/JP284047

Corresponding author(s): Martin Whitham (m.whitham@bham.ac.uk)

Review Timeline:

Submission Date:	01-Nov-2022
Editorial Decision:	07-Dec-2022
Revision Received:	25-Jan-2023
Accepted:	22-Feb-2023

Senior Editor: Harold Schultz

Reviewing Editor: Susan Currie

Transaction Report:

Dear Dr Whitham,

Re: JP-RP-2022-284047 "Single vesicle analysis reveals the release of tetraspanin positive extracellular vesicles into circulation with high intensity intermittent exercise" by Luke C McIlvenna, Hannah-Jade Parker, Alex P Seabright, Benedict Sale, Genevieve Anghileri, Samuel R.C Weaver, Samuel J.E. Lucas, and Martin Whitham

Thank you for submitting your manuscript to The Journal of Physiology. It has been assessed by a Reviewing Editor and by 2 expert referees and we are pleased to tell you that it is acceptable for publication following minor revision.

Please address all the points raised and incorporate all requested revisions or explain in your Response to Referees why a change has not been made. We hope you will find the comments helpful and that you will be able to return your revised manuscript within 12 weeks. If you require longer than this, please contact journal staff: jp@physoc.org.

REVISION CHECKLIST:

We look forward to receiving your revised submission.

Yours sincerely,

Harold D Schultz
Senior Editor
The Journal of Physiology
<https://jp.msubmit.net>
<http://jp.physoc.org>
The Physiological Society
Hodgkin Huxley House
30 Farringdon Lane
London, EC1R 3AW
UK
<http://www.physoc.org>
<http://journals.physoc.org>

REQUIRED ITEMS:

-Author photo and profile. First (or joint first) authors are asked to provide a short biography (no more than 100 words for one author or 150 words in total for joint first authors) and a portrait photograph. These should be uploaded and clearly labelled with the revised version of the manuscript. See Information for Authors for further details.

-You must start the Methods section with a paragraph headed Ethical Approval. If experiments were conducted on humans confirmation that informed consent was obtained, preferably in writing, that the studies conformed to the standards set by the latest revision of the Declaration of Helsinki, and that the procedures were approved by a properly constituted ethics committee, which should be named, must be included in the article file. If the research study was registered (clause 35 of the Declaration of Helsinki) the registration database should be indicated, otherwise the lack of registration should be noted as an exception (e.g. The study conformed to the standards set by the Declaration of Helsinki, except for registration in a database.). For further information see: <https://physoc.onlinelibrary.wiley.com/hub/human-experiments>

-Your manuscript must include a complete Additional Information section

-Please upload separate high-quality figure files via the submission form.

-A Statistical Summary Document, summarising the statistics presented in the manuscript, is required upon revision. It must be on the Journal's template, which can be downloaded from the link in the Statistical Summary Document section here: https://jp.msubmit.net/cgi-bin/main.plex?form_type=display_requirements#statistics

-Papers must comply with the Statistics Policy https://jp.msubmit.net/cgi-bin/main.plex?form_type=display_requirements#statistics

In summary:

-If n {less than or equal to} 30, all data points must be plotted in the figure in a way that reveals their range and distribution. A bar graph with data points overlaid, a box and whisker plot or a violin plot (preferably with data points included) are acceptable formats.

-If $n > 30$, then the entire raw dataset must be made available either as supporting information, or hosted on a not-for-profit

repository e.g. FigShare, with access details provided in the manuscript.

- 'n' clearly defined (e.g. x cells from y slices in z animals) in the Methods. Authors should be mindful of pseudoreplication.

- All relevant 'n' values must be clearly stated in the main text, figures and tables, and the Statistical Summary Document (required upon revision)

- The most appropriate summary statistic (e.g. mean or median and standard deviation) must be used. Standard Error of the Mean (SEM) alone is not permitted.

- Exact p values must be stated. Authors must not use 'greater than' or 'less than'. Exact p values must be stated to three significant figures even when 'no statistical significance' is claimed.

- Statistics Summary Document completed appropriately upon revision

- A Data Availability Statement is required for all papers reporting original data. This must be in the Additional Information section of the manuscript itself. It must have the paragraph heading "Data Availability Statement". All data supporting the results in the paper must be either: in the paper itself; uploaded as Supporting Information for Online Publication; or archived in an appropriate public repository. The statement needs to describe the availability or the absence of shared data. Authors must include in their Statement: a link to the repository they have used, or a statement that it is available as Supporting Information; reference the data in the appropriate section(s) of their manuscript; and cite the data they have shared in the References section. Whenever possible the scripts and other artefacts used to generate the analyses presented in the paper should also be publicly archived. If sharing data compromises ethical standards or legal requirements then authors are not expected to share it, but must note this in their Statement. For more information, see our Statistics Policy.

- Please include an Abstract Figure file, as well as the figure legend text within the main article file. The Abstract Figure is a piece of artwork designed to give readers an immediate understanding of the research and should summarise the main conclusions. If possible, the image should be easily 'readable' from left to right or top to bottom. It should show the physiological relevance of the manuscript so readers can assess the importance and content of its findings. Abstract Figures should not merely recapitulate other figures in the manuscript. Please try to keep the diagram as simple as possible and without superfluous information that may distract from the main conclusion(s). Abstract Figures must be provided by authors no later than the revised manuscript stage and should be uploaded as a separate file during online submission labelled as File Type 'Abstract Figure'. Please ensure that you include the figure legend in the main article file. All Abstract Figures should be created using BioRender. Authors should use The Journal's premium BioRender account to export high-resolution images. Details on how to use and access the premium account are included as part of this email.

EDITOR COMMENTS

Reviewing Editor:

While the reviewers have found this study of interest with potential for use in the field, they have raised a number of concerns and recommendations around use of the single vesicle analysis methodology. The reviewers feel that comparison with a more commonly used approach would be advised along with inclusion of analysis on EV depleted plasma to confirm lack of CD9, CD63 and CD81 in the plasma. Given that previous work has mostly used platelet free plasma, reviewers also feel it is important to characterise platelet poor plasma more fully. Taking all of the reviewers comments into account, a major revision of the paper is recommended.

Senior Editor:

Thank you for submission of your manuscript to The journal of Physiology. The manuscript will require a major revision based upon reviewer comments. Please note also the Journal policy for reporting p values and indicating statistical texts used with data comparisons.

Please show actual p values throughout, including within figures 1-5 (instead of asterisks). Please indicate in figure legends the statistical test used for p values, and for p values in the Results text.

REFEREE COMMENTS

Referee #1:

The manuscript by McIlvenna et al describes the analysis of tetraspanins, markers of small extracellular vesicles, in circulation of individuals pre and post high intensity interval training using a immunofluorescence microfluidic chip approach. With platelets known to act as a source of sEVs and therefore a potential confounder, the authors also assess the influence of platelets on plasma tetraspanins using two methods of centrifugation to reduce or deplete platelets from the plasma. This reviewer finds this study interesting and believes it is a useful study for the field. I am therefore supportive. However, I have a number of points that the authors may find useful to consider.

The introduction uses the description "small extracellular vesicles" which is an accurate term but to aid the reader perhaps the authors could consider quantifying "small", is this as is described in the MISEV 2018 criteria?

There is a relatively small number of female participants. A notation of limitation or caveat in the discussion is worth considering giving emerging data relating to the sexual dimorphism of sEV release and phenotype.

Given the increasing evidence for differential effects of exercise dependent on the time of exercise throughout the day (e.g. morning vs evening exercise), were all exercise interventions carried out at the same time of day?

To determine whether the single vesicle analysis method provides results of a comparative (or better) standard to more commonly (and established) methods for EV isolation (such as those the authors describe: polymer-based precipitation, ultrafiltration, differential centrifugation and size exclusion chromatography) the authors could consider a direct comparison to one of these.

The authors state: "underlying assumption in the presented finding is that the core tetraspanins we have analysed are on sEVs and not released into circulation independently. Since none of these proteins possess a signal peptide, this is unlikely." To confirm this, could the authors perform analysis on EV depleted plasma to confirm lack of CD9, CD63 and CD81 in the plasma.

minor:

Figure 2 A is introduced in the text prior to Fig 1.

Were ANOVA analyses subjected to a posthoc test (e.g. Dunnett's)?

Typo in the Discussion: "The core tetraspanins (CD9, C63..."

In the discussion, "however this is unlikely to be the main source." Since this has not been confirmed a potential suggestion would be "sole source" rather than "main source".

Referee #2:

The authors used single vesicle analysis to characterize the plasma sEV size and concentration in platelet poor and platelet-free fractions from human blood. It was found that for certain tetraspanin markers (CD9+ and CD41a), sEV size decreased slightly in PFP versus PPP samples, and this did not occur when analysing CD81+ or CD63+ targets. The addition of a secondary spin of plasma samples clearly decreased the presence of platelets, and the presence of a CD41a signal. In PFP samples, this also occurred in the presence of lower fluorescence for CD9, CD63, and CD81. To examine whether this technique would capture exercise-induced increases in plasma sEV, blood was collected from participants that had performed HIIT exercise. In both PPP and PFP samples, HIIT exercise increased the signal for CD9, CD63, and CD81, indicating both fractions using single vesicle analysis show similar effect of exercise on sEV secretion in terms of tetraspanin presence.

Overall, the study is interesting and conducted well. I do however have some concerns/questions that relate to the research impact and advancement.

1. The reduction in CD9 positive and CD41a particle size in the platelet free fraction - is this actually related to the platelet component producing sEV ex vivo or do most of the slightly larger EVs just happen to be collected during the first spin? Can the platelet activity be inhibited during sample collection to determine if platelets themselves in response to rest/exercise contribute to EV pool?

2. While the fluorescence of the various individual sEV markers in the PFP condition decreases compared to PPP, do the percentages of sEV containing 1 marker, 2, or all 3 EV markers differ between fractions? This initial baseline seems important in terms of evaluating the context of HIIT.

3. Beyond EV number, size, and perhaps proportion of common sEV markers found on sEV using the exoview, it would be interesting to examine whether common exercise-responsive proteins from previous screens occur in both PPP and PFP sEV, strengthening the notion (or not) that both can reproduce outcomes within the context of exercise.

4. In the results, it is indicated that the platelet-free fraction is the more representative fraction. It is unclear why this is the case based on figure 1 - does this just refer to the reduction in platelets? Please clarify.

5. With the HIIT exercise, there was no change in particle size in the platelet-free blood when evaluating CD9, CD63, CD81+ sEV, but there was an increase with the CD41a platelet marker and later the fluorescence of CD41a also appears increased but not quantified in the PFP fraction (Fig. 3C). What happened in the PPP fraction pre/post exercise, is everything related to platelet / markers increased across both fractions? Given that previous work mostly used PFP, it is important to characterize the PPP.

6. It is interesting that exercise effects on sEV secretion in terms of CD9, CD63, and CD81 fluorescence count is retained in both the PPP and PFP fractions. Given the different size distributions identified using the exoview tends to be smaller than other methods (e.g., ~60nm vs 100nm), and sensitivity varies, etc., are the exercise effects also maintained using exoview in PPP and PFP sEV isolated using more common techniques such as differential centrifugation? Do they retain similar size/morphology, tetraspanin composition? Combined with the current data, this could be really useful for the field and provide confidence in the use of either technique to study sEV size/number in the context of exercise, especially considering the latter can offer a greater opportunity for downstream analyses and functional assessments with the sEV.

END OF COMMENTS

Confidential Review

01-Nov-2022

We thank the editors and reviewers for their comments. We have included additional data and made edits to the manuscript which we feel has both improved it and addressed the concerns raised. A specific response to each comment is detailed below.

Senior Editor:

Please show actual p values throughout, including within figures 1-5 (instead of asterisks). Please indicate in figure legends the statistical test used for p values, and for p values in the Results text.

Thank you for highlighting this oversight. We have corrected the manuscript to align with the Journal's policy on reporting of statistics. Whilst reviewing the data we also spotted an error in the data presentation in the original Figure 1 which we have now corrected.

During the review period we were notified of a buyout of Nanoview biosciences by Unchained laboratories and we have therefore edited the method section to reflect this new supplier of the Exoview platform and consumables.

Reviewer 1

1) The introduction uses the description "small extracellular vesicles" which is an accurate term but to aid the reader perhaps the authors could consider quantifying "small", is this as is described in the MISEV 2018 criteria?

Many thanks for highlighting this. We have corrected this at the start of the manuscript for clarity.

2) There is a relatively small number of female participants. A notation of limitation or caveat in the discussion is worth considering giving emerging data relating to the sexual dimorphism of sEV release and phenotype

This is an interesting point. It is our interpretation of the literature that the emerging data describing a sexual dimorphism of EV release is largely associated with shedding vesicles >200nm in size. Hence, we did not take steps to stratify the data based on sex or include this as one of our original aims. However, we agree little is known regarding this and we have added a notation of limitation in the discussion, as suggested.

3) Given the increasing evidence for differential effects of exercise dependent on the time of exercise throughout the day (e.g. morning vs evening exercise), were all exercise interventions carried out at the same time of day?

Again, many thanks for pointing out this oversight, which we have corrected in the methods section.

- 4) *To determine whether the single vesicle analysis method provides results of a comparative (or better) standard to more commonly (and established) methods for EV isolation (such as those the authors describe: polymer-based precipitation, ultrafiltration, differential centrifugation and size exclusion chromatography) the authors could consider a direct comparison to one of these.*

Our position, which we hope we have got across in the manuscript, is that the method by which sEVs are isolated and analysed is guided by the downstream application and no single method is 'gold standard'. We are clearly endorsing the single EV method, but also highlight its limitations in the discussion section. Previous published works have used different methods, each with advantages and disadvantages and our main intention with this research was to build upon what has previously been shown, with a different approach, rather than carry out a direct comparison with other methods. Since we effectively arrive at the same conclusion, a much larger, comparative approach, using every other method could conceivably be unnecessary. What we feel our study adds is additional collective weight of evidence, allied to that performed using alternative approaches to support our main hypothesis that sEVs are released into circulation with exercise, which was our ultimate aim. We have edited small sections of the discussion section to clarify our position.

- 5) *The authors state: "underlying assumption in the presented finding is that the core tetraspanins we have analysed are on sEVs and not released into circulation independently. Since none of these proteins possess a signal peptide, this is unlikely." To confirm this, could the authors perform analysis on EV depleted plasma to confirm lack of CD9, CD63 and CD81 in the plasma.*

As mentioned in (4), a complete depletion of sEVs from plasma preps is challenging with no universal method available to achieve this. However, separating sEVs from plasma using a density gradient is one approach that several groups have carried out and clearly show a lack of presence of CD9, CD63 and CD81 in the non-EV fraction. We have included reference to these works in the manuscript to strengthen our argument that our data represent sEVs, rather than "free" tetraspanins in plasma.

Minor:

- 1) *Figure 2 A is introduced in the text prior to Fig 1.*

We agree this is unusual, but unavoidable to maintain the platelet depletion comparison data first and in its entirety in Figure 1.

- 2) *Were ANOVA analyses subjected to a posthoc test (e.g. Dunnett's)?*

Yes, thank you for highlighting this. We have included this detail in the methods.

- 3) *Typo in the Discussion: "The core tetraspanins (CD9, C63..."*

Thank you. This has been corrected.

4) *In the discussion, "however this is unlikely to be the main source." Since this has not been confirmed a potential suggestion would be "sole source" rather than "main source".*

We agree and have corrected this line in the manuscript.

Referee #2:

- 1) *The reduction in CD9 positive and CD41a particle size in the platelet free fraction - is this actually related to the platelet component producing sEV ex vivo or do most of the slightly larger EVs just happen to be collected during the first spin? Can the platelet activity be inhibited during sample collection to determine if platelets themselves in response to rest/exercise contribute to EV pool?*

As included in the discussion, we interpret our data that since there is a drop in platelet count with a double versus single spin and we observed an associated drop in tetraspanin positive sEV counts and protein expression, that the greater concentration in PPP is due to a contribution from the remnant platelets, *ex vivo*. This supports previous works using unbiased proteomic analyses, also mentioned in the existing discussion section. Simply, removing platelets from plasma reduces the sEV concentration. We have edited parts of the manuscript to make clear that this is our interpretation, supported by other published articles. We are not aware of any data implying large EVs can be pelleted at centrifugal speeds as low as 2500g that would offer an alternative explanation. As for the effect on sEV size, we are somewhat reluctant to overstate these data, since although statistically significant, a mean drop of 6-7nm, we feel is unlikely to be biologically meaningful.

- 2) *While the fluorescence of the various individual sEV markers in the PFP condition decreases compared to PPP, do the percentages of sEV containing 1 marker, 2, or all 3 EV markers differ between fractions? This initial baseline seems important in terms of evaluating the context of HIIT.*

We have carried out additional analyses of the PPP vs PFP to address this comment. Colocalisation data describes, as requested, the percentage of sEVs expressing each tetraspanin, in isolation and in combination, that contribute to the total captured sEV pool on each spot, in platelet poor and platelet free plasma. These data show (see below) there are modest changes but it's important to note that the number of sEVs in PFP is markedly lower (Figure 1). We have created drafts with this data included, but on proofreading the revised manuscript, we feel inclusion of these data actually complicates the overall message, that removal of remnant platelets from platelet poor plasma reduces the number of sEVs in the sample. We can, of course, include these data should the reviewer and editorial team feel it adds to the research. Considering the reviewers comments regarding percentage of sEVs expressing each marker the context of HIIT, we have provided a more in depth colocalisation analysis at baseline and post exercise in Figure 4.

3) *Beyond EV number, size, and perhaps proportion of common sEV markers found on sEV using the exoview, it would be interesting to examine whether common exercise-responsive proteins from previous screens occur in both PPP and PFP sEV, strengthening the notion (or not) that both can reproduce outcomes within the context of exercise.*

It's important to note that there is a distinction between sEV count and phenotype that we delineate in the discussion section. Part of our data support that the effect of aerobic exercise on sEV count is replicated in PPP and PFP. However, we cite more detailed, in-depth proteomic analyses that imply that if your research question is on the effect of exercise on the phenotype, or protein cargo of sEV, then platelet poor plasma is problematic. We wish to stress then, that our take home message is not so much that both PPP and PFP reproduce outcomes within the context of exercise, but either will likely suffice if your concern is sEV number. We agree wholeheartedly that a consideration of platelet contamination is important when interpreting all plasma work in the context of exercise. We are particularly reminded of the work of Matthias Mann's lab who show many potential biomarkers could be artifacts of remnant platelet contamination (Geyer PE, et al (2019). Plasma Proteome Profiling to detect and avoid sample-related biases in biomarker studies. *EMBO Molecular Medicine*; DOI: 10.15252/emmm.201910427.). However, as stated in the manuscript, our primary focus here is on sEVs and consideration of the effect of platelet depletion on other, non EV, exercise responsive secreted proteins, we feel is

somewhat beyond the scope of the article. To address any possible misinterpretation of our take home message regarding the suitability of platelet poor plasma in exercise studies, we have edited the manuscript throughout to provide clarity.

- 4) *In the results, it is indicated that the platelet-free fraction is the more representative fraction. It is unclear why this is the case based on figure 1 - does this just refer to the reduction in platelets? Please clarify.*

We interpret our data as support for previous works, using differing methods, that failure to remove platelets from plasma results in a higher sEV concentration in the sample. Since this implies an *ex vivo* contribution of sEV from remnant platelets, we feel this supports our conclusion that platelet free plasma is the most representative of circulating sEVs and discuss this at length in the discussion section. We have edited the manuscript throughout to make this clearer.

- 5) *With the HIIT exercise, there was no change in particle size in the platelet-free blood when evaluating CD9, CD63, CD81+ sEV, but there was an increase with the CD41a platelet marker and later the fluorescence of CD41a also appears increased but not quantified in the PFP fraction (Fig. 3C). What happened in the PPP fraction pre/post exercise, is everything related to platelet / markers increased across both fractions? Given that previous work mostly used PFP, it is important to characterize the PPP.*

It is important to note that the platform we have used does not supply Cd41a fluorescence data. Rather, CD41a antibodies are used to immobilise CD41a positive sEVs and the available fluorescent colour channels are taken by CD9, CD63 and CD81 antibodies – thought to be most representative of sEVs. As shown in Figure 1, we can, however assess the number of CD41a+ vesicles via interferometric counts, although we do provide a caveat in the methods section regarding its accuracy vs immunofluorescence. Despite this, and consistent with other markers, CD41a+ increased in response to exercise in platelet free plasma, which supports previous research suggesting a contribution of platelets *in vivo* to the sEV response to exercise (Brahmer et al 2020), although some data questions whether CD41a (ITGA2B) is specific to thrombocytes (Reickmann et al, 2017). We have included these data in Figure 3 and added some sentences to the discussion to describe our interpretation of it. We have also included interferometric counts of CD41a+ sEV in the PPP, as requested (Figure 5B). This also shows a significant increase with exercise, but consistent with point (4) of this rebuttal, and one of our take home messages, it is difficult to interpret these data because there is clearly a contribution of sEV from remnant platelets *ex vivo*.

- 6) *It is interesting that exercise effects on sEV secretion in terms of CD9, CD63, and CD81 fluorescence count is retained in both the PPP an PFP fractions. Given the different size distributions identified using the exoview tends to be smaller than other methods (e.g., ~60nm vs 100nm), and sensitivity varies, etc., are the exercise effects also maintained using exoview in PPP and PFP sEV isolated using more common techniques such as differential*

centrifugation? Do they retain similar size/morphology, tetraspanin composition? Combined with the current data, this could be really useful for the field and provide confidence in the use of either technique to study sEV size/number in the context of exercise, especially considering the latter can offer a greater opportunity for downstream analyses and functional assessments with the sEV.

Again, our position is that the method by which sEVs are isolated and analysed is guided by the downstream application and no single method is 'gold standard'. Previous works demonstrate sEVs are released into circulation with exercise using a range of different methods, each with advantages and disadvantages. Our primary aim was to further investigate this hypothesis with a novel and less invasive method, directly examining plasma. In doing so, we support the stated hypothesis and present evidence implying that any biases or artefact introduced by a pre-isolation step are unlikely to influence the overall conclusion that sEVs are released into circulation with aerobic exercise. UC isolated samples are prone to aggregation and represent a drastic concentration effect on the sample, both of which create issues for both the binding of the sEVs to the capture spots and the initial sample volume to use for a comparable analysis. It is known that several other common isolation methods result in a reduction of EV yield (counts and total protein) and recovery (Ter-Ovanesyan et al, (2021) Framework for rapid comparison of extracellular vesicle isolation methods. *Elife*, 10, p.e70725.) and these limitations are exactly what the Exoview approach looks to circumvent. So while we appreciate the suggestion of a direct comparison with sEVs pre-isolated with ultracentrifugation, if it were feasible, we are not convinced it adds useful data to test our hypothesis. That said, the Exoview approach doesn't inform morphology (as suggested) and other approaches are more suited to other applications, such as biological function, which we stress in an additional section of the discussion.

Dear Dr Whitham,

Re: JP-RP-2023-284047R1 "Single vesicle analysis reveals the release of tetraspanin positive extracellular vesicles into circulation with high intensity intermittent exercise" by Luke C McIlvenna, Hannah-Jade Parker, Alex P Seabright, Benedict Sale, Genevieve Anghileri, Samuel R.C Weaver, Samuel J.E. Lucas, and Martin Whitham

We are pleased to tell you that your paper has been accepted for publication in The Journal of Physiology.

Authors should note that it is too late at this point to offer corrections prior to proofing. The accepted version will be published online, ahead of the copy edited and typeset version being made available. Major corrections at proof stage, such as changes to figures, will be referred to the Editors for approval before they can be incorporated. Only minor changes, such as to style and consistency, should be made at proof stage. Changes that need to be made after proof stage will usually require a formal correction notice.

Yours sincerely,

Harold D Schultz
Senior Editor
The Journal of Physiology
<https://jp.msubmit.net>
<http://jp.physoc.org>
The Physiological Society
Hodgkin Huxley House
30 Farringdon Lane
London, EC1R 3AW
UK
<http://www.physoc.org>
<http://journals.physoc.org>

P.S. - You can help your research get the attention it deserves! Check out Wiley's free Promotion Guide for best-practice recommendations for promoting your work at www.wileyauthors.com/eeo/guide. You can learn more about Wiley Editing Services which offers professional video, design, and writing services to create shareable video abstracts, infographics, conference posters, lay summaries, and research news stories for your research at www.wileyauthors.com/eeo/promotion.

IMPORTANT NOTICE ABOUT OPEN ACCESS: To assist authors whose funding agencies mandate public access to published research findings sooner than 12 months after publication, The Journal of Physiology allows authors to pay an Open Access (OA) fee to have their papers made freely available immediately on publication.

You can check if your funder or institution has a Wiley Open Access Account here: <https://authorservices.wiley.com/author-resources/Journal-Authors/licensing-and-open-access/open-access/author-compliance-tool.html>.

EDITOR COMMENTS

Reviewing Editor:

All of the reviewers previous concerns have now been addressed and the manuscript is acceptable for publication.

Senior Editor:

Thank you for submission of your manuscript to the Journal of Physiology. We are pleased to announce that the revised manuscript is now appropriate for publication in the journal. Please consider our journal for our future efforts.

REFEREE COMMENTS

Referee #1:

The authors have addressed my comments. I have no further points.

Referee #2:

The authors have addressed my concerns. I have no further comments.

1st Confidential Review

25-Jan-2023